biological reaction kinetics; biomolecular systems; diffusion; DNA; dynamics and function; helicases; molecular machines; single-molecule; random walk; magnetic and optical tweezers

**Corresponding author:**
Maria Manosas;
Email: mmanosas@gmail.com

CAMBRIDGE
UNIVERSITY PRESS

# Continuous-time random walk model for the diffusive motion of helicases

Victor Rodríguez-Franco[1] [iD], Michelle Marie Spiering[2], Piero Bianco[3], Felix Ritort[1,4,5] and Maria Manosas[1,4]

[1]Small Biosystems Lab, Departament de Física de la Matèria Condensada, Facultat de Física, Universitat de Barcelona, Carrer de Martí i Franquès, 1, 08028 Barcelona, Spain; [2]Department of Chemistry, The Pennsylvania State University, University Park, PA, USA; [3]Department of Pharmaceutical Sciences, College of Pharmacy, University of Nebraska Medical Center, Omaha, NB, USA; [4]Institut de Nanociència i Nanotecnologia, Universitat de Barcelona, Barcelona, Spain and [5]Reial Acadèmia de Ciències i Arts de Barcelona (RACAB), Barcelona, Spain

## Abstract

DNA helicases are molecular motors that use the energy from nucleotide hydrolysis to move along DNA, promoting the unwinding or rewinding of the double helix. Here, we use magnetic and optical tweezers to track the motion of three helicases, gp41, RecQ, and RecG, while they unwind or rewind a DNA hairpin. Their activity is characterized by measuring the helicase velocity and diffusivity under different force and ATP conditions. We use a continuous-time random walk framework that allows us to compute the mean helicase displacement and its fluctuations analytically. Fitting the model to the measured helicase velocity and diffusivity allows us to determine the main states and transitions in the helicase mechanochemical cycle. A general feature for all helicases is the need to incorporate an off-pathway pausing state to reproduce the data, raising the question of whether pauses play a regulatory role. Diffusivity measurements also lead to estimations of the thermodynamic uncertainty factor related to the motor efficiency. Assuming a tight mechano-chemical coupling, we find that the RecG helicase reaches a high efficiency when operating uphill, whereas the unwinding gp41 and RecQ helicases display much lower efficiencies. Incorporating the analysis of fluctuations allows for better characterization of the activity of molecular machines, which represents an advance in the field.

## Introduction

Helicases are ubiquitous enzymes that are present in all living organisms. They work as molecular motors, converting the chemical energy from the hydrolysis of adenosine triphosphate (ATP) into mechanical work and motion along nucleic acids (NA) (Lohman and Bjornson, 1996; Tuteja and Tuteja, 2004). Helicase motion is coupled to various functions, such as NA unwinding and separation of the two strands of the duplex (e.g. in DNA replication and transcription); NA rewinding and formation of multi-branch structures (e.g. in DNA recombination); and disruption of protein-NA interactions (e.g. in DNA repair and RNA processing) (Tuteja and Tuteja, 2004; Delagoutte and Von Hippel, 2003; Pyle, 2008). These diverse functions are accomplished by the presence of different helicases, classified into families based on their conserved structural motifs (Caruthers and McKay, 2002; Singleton *et al.*, 2007). Besides their relevance in molecular biology, helicases, and molecular motors in general have raised interest in many other fields, and in physics, they are a paradigm of small systems (Bustamante *et al.*, 2005). Molecular motors typically move in nanometer (nm) steps while generating mechanical forces in the picoNewton (pN) range, producing work of pN·nm, which is on the order of the thermal energy unit $k_B T$ ($1 k_B T = 4$pN·nm). On the other hand, the energy released by the hydrolysis of ATP is $\sim 10 k_B T$, which implies that these motors work in a strong Brownian environment where fluctuations play a central role.

The helicase translocation along the NA strand in a defined directionality ($3'$ to $5'$ or $5'$ to $3'$) results from different nucleotide-enzyme conformations of distinct NA affinities connected in a cyclic reaction network (Delagoutte and Von Hippel, 2002; Patel and Donmez, 2006). The mechano-chemical coupling between the ATP hydrolysis reaction and the enzyme motion can be described in terms of two mechanisms: the Brownian Ratchet (BR) and the Power Stroke (PS) (Pyle, 2008; Patel and Donmez, 2006; Parrondo and Español, 1996; Hwang and Karplus, 2019; Galburt and Tomko, 2017). These two mechanisms exemplify the two limit cases: biasing or rectifying the motor diffusive thermal motion (BR) versus downhill dynamics based on a motor structural change induced by ATP hydrolysis (PS). In practice, many molecular motors may employ a combination of both mechanisms to achieve efficient motion (Hwang and Karplus, 2019). These two mechanisms can also be used to describe motor translocation coupled to NA strand separation/rewinding during NA unwinding/rewinding, leading to the classification of helicases as active (for PS) and passive (for BR) (Patel and Donmez, 2006; Galburt and Tomko, 2017; Manosas *et al.*, 2010). The active helicase directly interacts with the ssNA/dsNA junction,

destabilizing (or stabilizing) the base-pairs (bp) at the fork before translocation. In contrast, the passive helicase relies on the thermal fraying of the bp at the NA fork to promote un(re)winding.

Traditionally, helicase activity has been characterized using ensemble assays, such as gel-based or fluorescence spectroscopy (Matson *et al.*, 1983; Raney *et al.*, 1994; Kim and Seo, 2009). These approaches measure average properties over a large ensemble of molecules, but they provide limited information about nanoscale processes where fluctuations are relevant. In the last 30 years, different single-molecule techniques have emerged (Neuman and Nagy, 2008; Joo *et al.*, 2008), allowing for monitoring the activity of individual enzymes in real-time. These measurements facilitate the detection of molecular heterogeneity, rare events, pathways, intermediates, and dynamical NA-helicase interactions (Yodh *et al.*, 2010), crucial aspects for understanding how helicases work. The movement of single helicases can be monitored using single-molecule force spectroscopy techniques, such as optical, magnetic, or nanopore tweezers (Neuman and Nagy, 2008; Cheng *et al.*, 2007; Hormeno *et al.*, 2022; Craig *et al.*, 2017). In these assays, a mechanical force is applied on the enzyme or along the NA substrate, and the NA extension along the force direction is measured, defining a reaction coordinate to follow the advance of the helicase (Tinoco and Bustamante, 2002). By modifying the nucleotide conditions (e.g. ATP and ADP concentrations) and the applied force, the coupling between the helicase motion and the ATP hydrolysis reaction can be investigated, discriminating between different helicase mechanisms (Manosas *et al.*, 2010; Dumont *et al.*, 2006; Ribeck *et al.*, 2010; Spies, 2014; Seol *et al.*, 2019; Laszlo *et al.*, 2022). The general approach has been to use the experimental data to test specific models for each helicase, finding that, in many cases, different helicases display distinct activities involving complex reactions with multiple kinetic pathways and/or different rate-limiting steps.

Most descriptions of molecular motors are based on simple kinetic models and pathways. From muscle transport motors such as kinesin and myosin, to genomic maintenance machines such as polymerases and helicases, most models have focused on describing the average motor velocity under different conditions of ATP, ADP, force, and temperature (Kolomeisky and Fisher, 2007; Astumian, 2010; Keller and Bustamante, 2000). However, a main feature of these machines is the Brownian fluctuations and their diffusivity. Despite its importance, the helicase diffusivity has not been analysed in detail before. The diffusion constant ($D$), along with the velocity ($v$), are of particular interest as they are related to the randomness parameter $r = \frac{2D}{dv}$, where $d$ is the motor step-size (Svoboda *et al.*, 1994). This parameter provides information about the number of rate-limiting steps in an enzymatic cycle, being $r = 1$ for a Poisson process and $r = 0$ for a molecular clock. Alternatively, the motor step-size $d$ can be estimated if the number of rate-limiting steps is known (Neuman *et al.*, 2005). Additionally, $v$ and $D$ are related to the $Q$ factor of the thermodynamic uncertainty relation (TUR) (Barato and Seifert, 2015; Song and Hyeon, 2021). The TUR sets an inequality between the entropy production rate $\sigma$ and the measurement precision in nonequilibrium steady states. It is defined in terms of generic nonequilibrium currents, which for the case of a translocating motor takes the form $Q = \sigma \frac{2D}{k_B v^2} \geq 2$ with $\sigma$ expressed in $k_B$ units (Song and Hyeon, 2021). This factor quantifies the irreversibility of an enzymatic process in the nonlinear regime and is related to the motor's thermodynamic efficiency $\eta$ defined as the ratio between the amount of delivered mechanical work ($W$) and the input chemical energy coming from the ATP hydrolysis ($\Delta\mu$): $\eta = \frac{W}{\Delta\mu}$. The two quantities satisfy the energy balance relation, $\Delta\mu = W + T\sigma t$, where $T$ is the temperature and $T\sigma$ stands for the heat rate released to the environment. The thermodynamic efficiency $\eta$ can be expressed in terms of $Q$, $\eta = \left(1 + Q \frac{vd}{2D} \frac{k_B T}{W}\right)^{-1}$ (Song and Hyeon, 2021; Pietzonka *et al.*, 2016), small $Q$ values indicating a more efficient motor that operates closer to the limits of thermodynamic optimization.

In this work, we have investigated three different DNA helicases using magnetic tweezers (MT) and optical tweezers (OT). Two of them catalyse DNA unwinding: the T4 gp41 helicase, which is involved in DNA replication in T4 bacteriophage, and RecQ from *Escherichia Coli* (*E. coli*), which participates in different DNA repair pathways. The third one is the RecG helicase from *E. coli*, which is involved in DNA repair and recombination, catalysing DNA rewinding and the formation of multi-branched DNA structures (Manosas *et al.*, 2010; Venkatesan *et al.*, 1982; Manosas *et al.*, 2013; Lionnet *et al.*, 2007; Bagchi *et al.*, 2018; Umezu *et al.*, 1990; McGlynn and Lloyd, 2001). In the experiments, a constant force is applied to the extremities of a DNA hairpin, and the helicase un(re)winding activity is followed by measuring the changes in the DNA extension, enabling real-time monitoring of the enzyme activity. From these measurements, we can infer the position of the helicase along the DNA and extract its mean velocity, diffusivity, and pause kinetics. To interpret the experimental results, we have developed a general theoretical framework for helicase motion based on random-walk theory. The model features a continuous-time random walk (CTRW) in a one-dimensional chain with an auxiliary pausing state. There are three distinct kinetic transitions: forward, backward, and enter/exit a pause. Each transition is described by activated kinetic rates that depend on the force and ATP concentration ([ATP]). The model fits the translocation and unwinding/rewinding rates and diffusivity data for the different helicases over a range of forces and [ATP]. The fitting procedure allows us to determine the chemical (ATP concentration) and mechanical (force) dependencies of the kinetic rates connecting the different states, providing insight into the helicase mechano-chemical cycle. Assuming a tight mechano-chemical coupling, we have also investigated how the $Q$ factor and the motor efficiency $\eta$ change for each helicase and how their values depend on the helicase step-size and its active and passive nature.

## Methods

### DNA substrates

Two different hairpins, h1.2 and h1.4, of ∼1.2 and ∼1.4 kbp stems, respectively, were used for the MT and OT assays. Both hairpins have a 5′ single-stranded DNA (ssDNA) tail of ∼80 nucleotides (nts) labelled with a biotin and 3′ ssDNA tail of ∼100 nts labelled with several digoxigenins. The h1.2 hairpin was prepared as previously described in (Manosas *et al.*, 2009). The stem of the h1.4 hairpin comes from a segment of the plasmid PBR322 and four different oligonucleotides are annealed and ligated to generate the loop and handles (details in Supplementary Figure S1). Experiments with *E. coli* RecQ were performed with the h1.2 hairpin and experiments with RecG and gp41 were performed with the h1.4 hairpin.

### Enzyme preparation and experimental conditions

The different helicases, T4 gp41, *E. coli* RecQ and *E. coli* RecG, were purified as previously described (Valentine *et al.*, 2001; Bernstein

and Keck, 2003; Slocum *et al.*, 2007). In this study, we used a truncated version of the RecQ helicase lacking the HRDC (Helicase and RNase D C-terminal) domain, referred to as RecQ-Δ in previous studies (Bagchi *et al.*, 2018; Bernstein and Keck, 2003). Hereafter, we will simply use the notation RecQ to refer to the RecQ-Δ variant.

Assays with gp41 and RecG were performed in a buffer containing 25 mM Tris–Ac (pH 7.5), 10 mM Mg(OAc)$_2$, 150 mM KOAc, 1 mM dithiothreitol (DTT), and different ATP concentrations (0.5–4 mM for gp41 and 100 μM-2 mM for RecG). Assays with RecQ were performed in a buffer containing 20 mM Tris–HCl (pH 7.5), 25 mM NaCl, 3 mM Mg(Cl)$_2$, 1 mM DTT and different ATP concentrations (40 μM to 1 mM). All experiments were done at 25 °C. The protein concentration was 50 nM for gp41 (monomeric concentration), 10 nM for RecG, and 30 pM for RecQ. These concentrations were chosen to optimize single-molecule conditions. To do so, we checked that the mean duration of a single unwinding trace (typically from a few seconds to a few tens of seconds, depending on the helicase, the ATP concentration, and the force conditions) is much smaller than the time between events, typically by a factor ∼10. This condition minimizes the probability of events where multiple helicases translocate simultaneously on the same hairpin.

## Single-molecule experiments

In magnetic tweezers (MT) experiments, we use a PicoTwist MT instrument (www.picotwist.com) to manipulate DNA hairpins tethered between a micrometric magnetic bead and the glass surface of a microfluidic chamber (Figure 1a). For making the tethers the glass surface is treated with an anti-digoxigenin antibody and passivated with bovine serum albumin and the micron-sized magnetic beads are coated with streptavidin (Invitrogen MyOne). The applied force is controlled by adjusting the distance between the magnets and the sample ($Z_{mag}$). The microfluidic chamber is illuminated by a red LED that generates a parallel and monochromatic illumination. Using an inverted microscope connected to a CMOS camera we image the beads. The images are decorated by a set of diffraction rings, enabling real-time tracking of beads' 3D position with nanometric resolution at 30–80 Hz (Gosse and Croquette, 2002; Lionnet *et al.*, 2012). From the bead's $z$ position we obtain the extension of the DNA molecule. The bead's fluctuations in the $x - y$ plane are used to measure the force via the equipartition theorem (Gosse and Croquette, 2002; Lionnet *et al.*, 2012). An average calibration curve $F(Z_{mag})$ is used to estimate the force with 10% error due to bead inhomogeneities

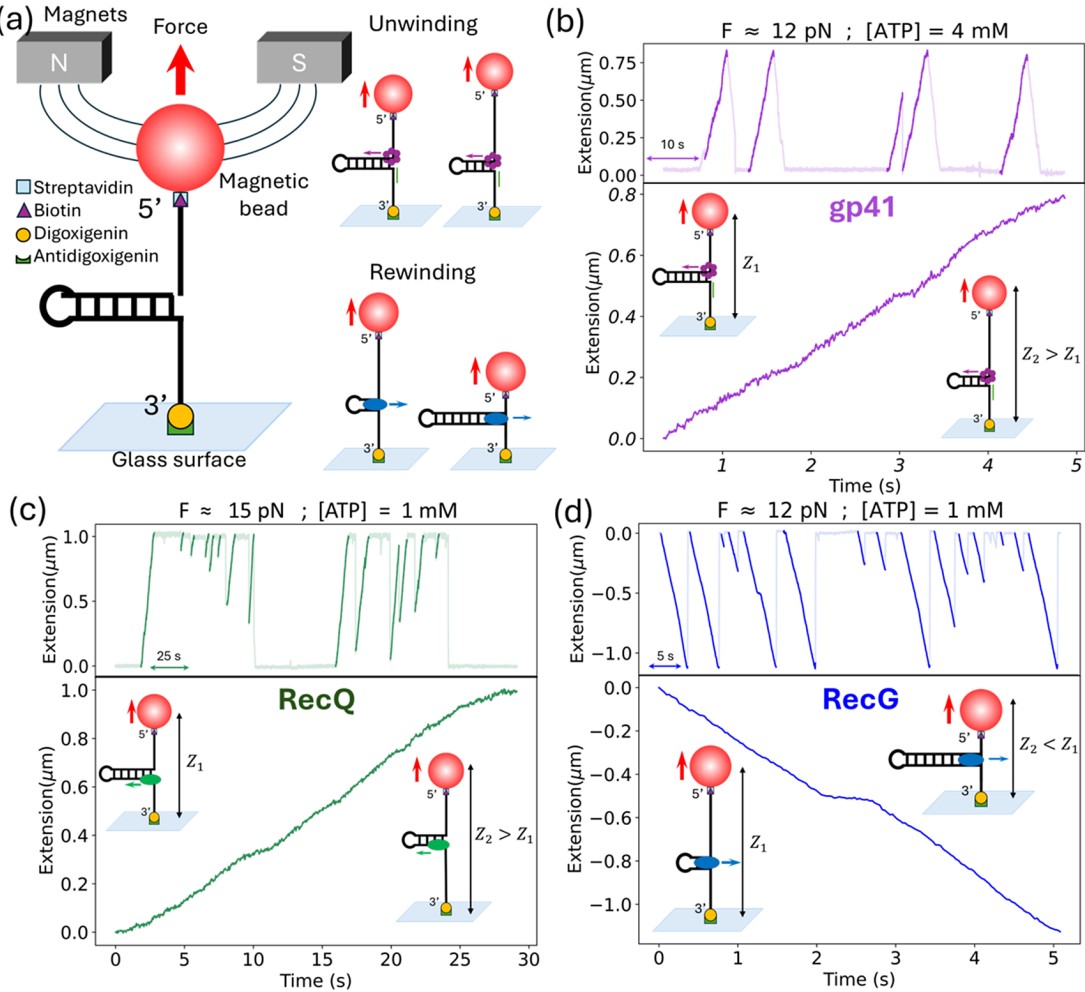

**Figure 1.** Helicase experiments (a) Schematic representation of the MT experimental setup, where a DNA hairpin is tethered between a glass surface and a magnetic bead. The progress of the unwinding and rewinding reactions leads to changes in the molecule extension. (b–d) Top panel: Experimental traces showing the hairpin extension as a function of time for gp41 (purple), RecQ (green), and RecG (blue) helicases. The un(re)winding events are highlighted in dark colour. Bottom panel: Details of a single un(re)winding trace with the schematic representations of the hairpin state at the beginning (left) and at the middle (right) of the trace.

(Lionnet *et al.*, 2012). In OT experiments, the DNA molecule is tethered between two micron-sized polystyrene beads using biotin–streptavidin and digoxigenin–antidigoxinein bonds. One bead is immobilized on the tip of a micropipette and the second bead is captured in an optical trap generated by two counter-propagating lasers (Supplementary Figure S4a). The force acting on the bead can be measured from the change in light momentum deflected by the bead using position-sensitive detectors (Smith *et al.*, 2003). After injecting the helicases and ATP, unwinding and rewinding activities are detected as an increase and decrease of the measured DNA hairpin extension respectively (Figure 1 and Supplementary Figure S4).

MT are used to test the activity of helicases in a force range going from 5 to 15 pN. Typically, about 50–100 beads are tracked simultaneously, which allows us to obtain large statistics. For the unwinding helicases (gp41 and RecQ), we started with the hairpin formed at a force below the unzipping force (15 pN) and monitored the unwinding activity from the DNA extension changes (Figure 1b,c). The gp41 helicase requires a long 5′ tail for efficient loading (Valentine *et al.*, 2001; Feng *et al.*, 2023). For this reason, we used a 40-mer oligonucleotide that is complementary to a hairpin region located at ∼500 bp from the fork. By unzipping the hairpin (at $F > 15$ pN) we hybridized the oligonucleotide, generating a ∼500 nt 5′ and 3′ tails (Supplementary Figure S2).

For the RecG rewinding helicase, we started with a partially unzipped hairpin. The partially unzipped configuration was achieved by either hybridizing a short oligonucleotide complementary to a hairpin region or by using forces close to the unzipping force. In the former case, the force was first increased to mechanically unzip the hairpin and allow the oligonucleotide to bind. Next, the force was decreased to a given value (typically between 5 and 13 pN) and the hairpin partially reforms until reaching the oligonucleotide that blocks the full re-zipping (Supplementary Figure S3a). In the latter case, the force was set to a value (∼ 15 pN) where the hairpin unzips except the last ∼50 bases containing a GC-rich region that requires larger forces to unzip (Supplementary Figure S3b). In both cases, the rewinding of the partially unzipped hairpin was detected as a decrease in the measured DNA extension (Figure 1 and Results).

OT were also used to test the activity of the RecG helicase at larger forces (above 15 pN). In OT assays, we initially increased the distance between the micropipette and the trap ($X_T$) reaching a force of ∼15 pN and a partially unzipped hairpin. In the presence of RecG, the rewinding reaction causes a shortening of the DNA that induces the displacement of the bead in the trap, generating an increase in force (Supplementary Figure S4b). By using a force feedback protocol we can test the rewinding activity at different forces. We did not observe any activity above ∼35 pN, in agreement with previous measurements (Manosas *et al.*, 2013). These large forces probably induce the stalling and dissociation of the enzyme from the DNA template.

## CTRW model

The movement of the helicase along DNA can be modelled as a one-dimensional random walk on a chain, where the walker can perform different transitions (Figure 2a): forward movement with probability $P_+$ and step $d_+$, backward movement with probability $P_-$ and step $d_-$ and pausing with probability $P_0$ and step $d = 0$. These probabilities satisfy the normalization condition: $P_+ + P_- + P_0 = 1$. Each transition is governed by an exponentially distributed intrinsic time with an average value: $\tau_+$ for the forward transition, $\tau_-$ for the backward transition, and $\tau_0$ for the pause. This kinetic scheme can be described within the continuous-time random walk (CTRW) framework (Wang *et al.*, 2020; Kutner and Masoliver, 2017). In this formalism, the walker dynamics is described using two stochastic distributions: the jump distribution $f(x)$, which represents the probability of a displacement $x$ during a single step and depends on the probabilities $P_+$, $P_-$ and $P_0$, and the waiting time distribution $\psi(t)$, which describes the time interval $t$ before a transition occurs and depends on the intrinsic times $\tau_+, \tau_-,$ and $\tau_0$. The probability of locating the walker at a distance $x$ at a time $t$, $P(x,t)$, is given by the Montroll–Weiss expression in Fourier–Laplace space ($s$ is the Laplace transform of $t$ and $k$ the Fourier transform of $x$) (Wang *et al.*, 2020; Montroll and Weiss, 1965),

$$P(k,s) = \frac{1 - \hat{\psi}(s)}{s} \frac{1}{1 - \hat{\phi}(k,s)}, \tag{1}$$

where $\hat{\psi}(s)$ is the Laplace transform of the waiting time distribution $\psi(t) = \sum_{i=+,-,0} \frac{1}{\tau_i} e^{-t/\tau_i}$ and $\hat{\phi}(k,s)$ is the Fourier–Laplace transform of the one-step joint distribution $\phi(x,t)$. Taking into account that the waiting time and jumps are correlated: $\phi(x,t) = \psi(t)f(x|t)$, being $f(x|t)$ the conditional probability density function to have a displacement $x$ in a time interval $t$ during a single step. The one-step joint distribution reads as:

$$\begin{aligned} \phi(x,t) &= \frac{P_-}{\tau_-} e^{-t/\tau_-} \delta(x + d_-) \\ &+ \frac{P_0}{\tau_0} e^{-t/\tau_0} \delta(x) \\ &+ \frac{P_+}{\tau_+} e^{-t/\tau_+} \delta(x - d_+). \end{aligned} \tag{2}$$

Equation (1) can be analytically solved in the long-time regime ($k \to 0, s \to 0$) giving (Supplementary Section V):

$$P(k,s) = \frac{1}{s\left[1 - ik\frac{a_1}{sE} - ik\frac{a_2 E - a_1 F}{E^2} + \frac{k^2}{2}\frac{c_1}{sE}\right]}, \tag{3}$$

$$\begin{aligned} \text{with} \quad a_1 &= P_+ d_+ - P_- d_-, \\ a_2 &= P_+ d_+ \tau_+ - P_- d_- \tau_-, \\ c_1 &= P_+ d_+^2 + P_- d_-^2, \\ E &= P_- \tau_- + P_0 \tau_0 + P_+ \tau_+, \\ F &= P_- \tau_-^2 + P_0 \tau_0^2 + P_+ \tau_+^2. \end{aligned}$$

Performing the inverse Laplace–Fourier transform, we obtain a Gaussian distribution of displacements and times with an average velocity $v$ and a diffusivity $D$ given by:

$$v = \frac{a_1}{E}, \quad D = \frac{c_1}{2E} + \frac{a_1}{2E^2}(a_2 E - a_1 F). \tag{4}$$

Details of the model are presented in Supplementary Section V.

## Data analysis

The DNA extension as a function of time measured in MT (Figure 1b–d upper panels) and OT experiments (Supplementary Figure S4b) have been converted from nm to bp by using a conversion factor that depends on the force applied. This factor is

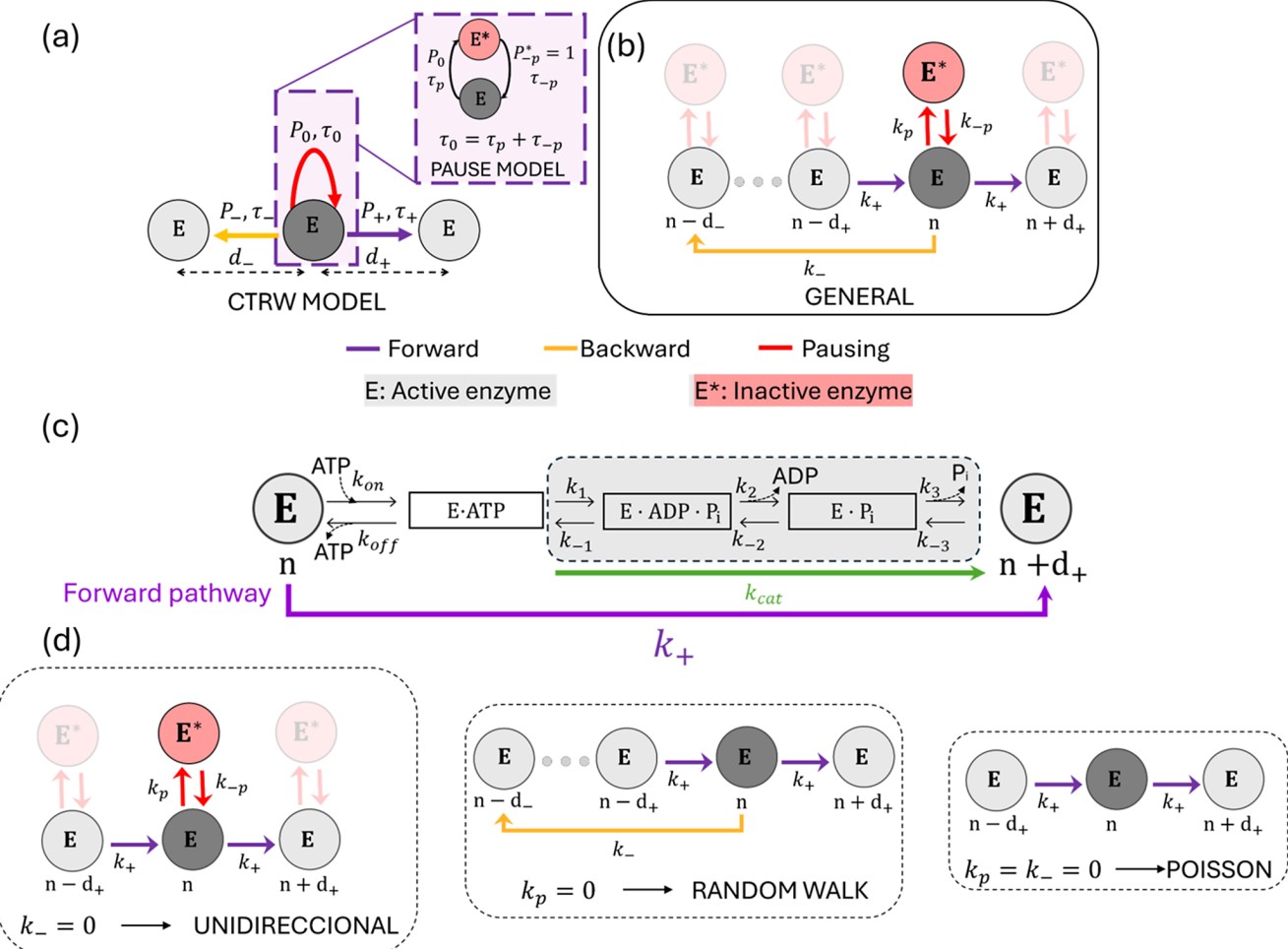

**Figure 2.** Helicase models. (a) Diagram of the CTRW model used to describe helicase motion. Inset shows the diagram of the pause transition divided into two steps corresponding to entering and exiting the pause state. $E$ and $E^*$ represent the states of a translocation-active and a pause-inactive helicase (b) Diagram of the general model with kinetic rates, which includes three pathways: the forward ATP-hydrolysis coupled to translocation on-pathway transition (purple), the off-pathway backward transition (yellow), and the off-pathway pausing transition (red). Kinetic rates and distances between states are indicated for each transition. (c) Details of the forward ATP-hydrolysis coupled to the translocation pathway indicating the intermediate steps. The overall rate $k_+$ (purple) integrates the ATP binding and unbinding reaction as well as the ATP hydrolysis and release of ADP and Pi. The ATP hydrolysis reaction (grey shaded area) is highly irreversible and can be approximated with a single kinetic rate $k_{cat}$ (green). (d) Sub-models derived from the general model: the unidirectional model (without backward motion, $k_- = 0$), the random walk (absence of off-pathway pausing state, $k_p = 0$) and the Poisson model ($k_- = 0$ and $k_p = 0$).

obtained from the elastic properties of the ssDNA molecule (Viader-Godoy *et al.*, 2021) as described in Supplementary Figure S5. In MT experiments, multiple tethers could be tracked simultaneously in a single experiment ($\sim 50$), each tether typically exhibiting $\sim 10$–50 unwinding/rewinding traces. In OT, different tethers were tested successively. Typically, we used about 3–5 tethers, each presenting $\sim 5$–20 traces. For each tether (in MT or OT), we computed the average velocity $v$ and the diffusivity $D$. The velocity was determined as the slope of the linear fit of the mean displacement over time, $\langle \Delta x \rangle = vt$ (top insets Figure 3a,c,e). The diffusivity was determined as half of the slope of the linear fit of the mean square displacement (MSD) over time, $\langle \Delta x^2 \rangle = 2Dt$ (bottom insets Figure 3a,c,e). The MSD presented an initial regime (at very low times) that deviates from the linear behaviour due to the Brownian motion of the bead (Supplementary Figure S6) (Neuman *et al.*, 2005). We then performed the fits with a time offset of few milliseconds.

We also analysed the pauses along the experimental traces by using a pause detection algorithm based on change point detection (Truong *et al.*, 2020) (Figure 3b,d,f and Supplementary Section VI). The algorithm depends on several parameters that are chosen based on the intrinsic noise level of the experimental traces (Supplementary Section VI). Nevertheless, sensitivity to short or long pauses depends strongly on the chosen parameters: optimizing for short pauses can fragment long ones, while detecting long pauses can lead to missing short events. To address this challenge, we optimized the parameters to reliably detect long pauses and analysed only those pauses longer than a threshold time $\tau_s$. Assuming that pause durations follow an exponential distribution, we fitted the histogram of detected pauses above $\tau_s$ to extract the characteristic pause time. The threshold $\tau_s$ was determined as the point where the histogram begins to deviate from exponential behaviour and typically corresponds to a short time on the order of the second or a fraction of a second. This method is illustrated in Supplementary Figure S9. To further validate this procedure, we simulated the continuous-time random walk (CTRW) model with the parameters shown in Table 3 and using the pause detection and histogram fitting protocol, we successfully recovered the

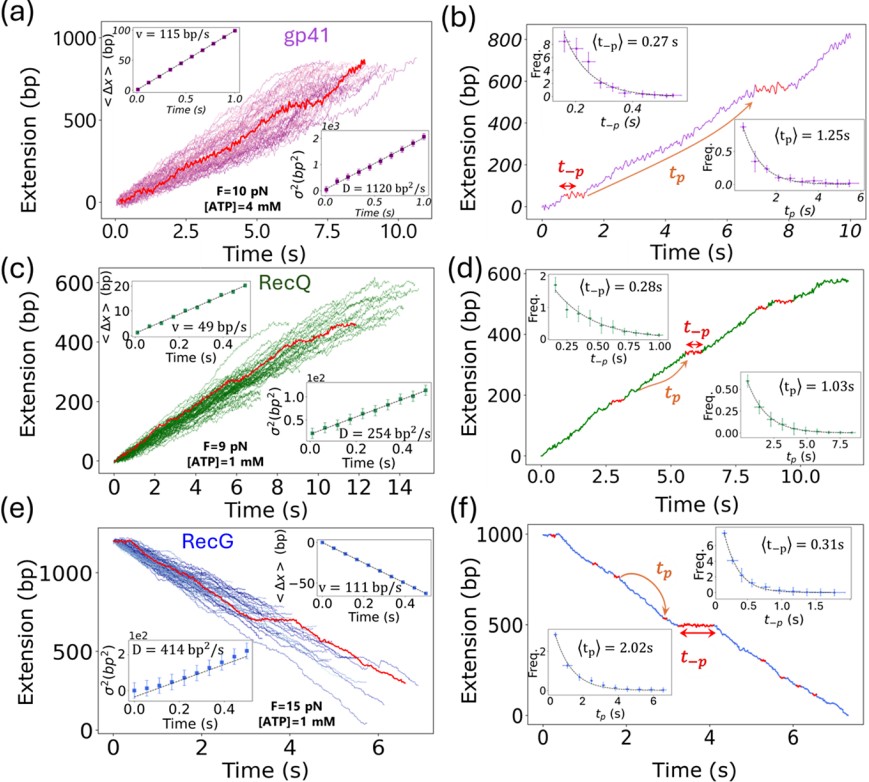

**Figure 3.** Helicase velocity, diffusivity, and pause kinetics. (a,c,e) Set of experimental traces ($\sim 50$) showing the DNA extension in bps as a function of time for gp41 (purple), RecQ (green), and RecG (blue). A single trace is shown in red as an example. Insets show the mean and variance of the helicase displacement as a function of time, computed from all traces in the main plot. Linear fits are shown as solid lines. Error bars are the standard error of the mean. (b,d,f) A single experimental trace for gp41 (purple), RecQ (green), and RecG (blue), showing the pauses detected with the step-finding algorithm in red. Insets show the distribution of the pause time $t_{-p}$ (top) and the time between pauses $t_p$ (bottom) from all the traces in panels (a), (c), and (e). Exponential fits are shown as continuous lines. Error bars are estimated using the bootstrap method.

theoretical predictions of our model (Eq. 7) within $\sim 5\%$ accuracy (Supplementary Section VI and Figure S11). We further checked the robustness of the pause detection analysis by tuning the algorithm parameters (Supplementary Section VI) and using an alternative velocity analysis (Supplementary Figures S12, S13, and S14). The experimental histogram of pausing times fitted to an exponential distribution is used to obtain the characteristic time to exit the pause, $\langle t_{-p} \rangle = \frac{1}{k_{-p}}$ (top insets Figure 3b,d,f). The time lag between pauses also follows an exponential distribution (bottom insets Figure 3b,d,f), yielding a value for the characteristic time to enter the pause $\langle t_p \rangle = \frac{1}{k_p}$. For each helicase, the velocity $v$, diffusivity $D$ and pause kinetics, $k_{-p}$ and $k_p$, at each experimental condition ([ATP] and force) were computed as the average value between different tethers, typically $\sim$10–50 tethers.

## Results

To investigate the activity of helicases, we used magnetic and optical traps to mechanically manipulate a DNA hairpin while monitoring the changes in DNA extension as the helicase unwinds or rewinds the hairpin. In MT experiments, the DNA hairpin was tethered between a glass surface and a micron-sized magnetic bead, and force was applied using a couple of magnets located on top of the microfluidic chamber (Figure 1a). In OT experiments, the DNA hairpin was tethered between two micron-sized beads, one held in an optical trap and the other fixed on the tip of a micro-pipette (Supplementary Figure S4). In both assays, a mechanical force was applied to destabilize the DNA hairpin duplex, serving as a means

to either assist the advance of a DNA unwinding helicase or hinder the rewinding activity of a DNA rewinding helicase. Details of the single-molecule experiments are presented in Section *Single-molecule experiments* in Methods.

### *Measuring DNA unwinding and rewinding activities*

A force above $\sim 15$ pN mechanically unzips the hairpin (Supplementary Figure S2). Below $\sim$15 pN, where the hairpin is mechanically stable, we can monitor the DNA unwinding catalysed by helicases that was detected as a smooth increase in the measured extension (Figure 1b,c). Unwinding activity was measured using MT to track different DNA hairpins in parallel, increasing the statistics. Here, we studied two different unwinding helicases: gp41 and RecQ. The gp41 is a hexameric helicase from the T4 bacteriophage that promotes DNA unwinding during DNA replication (Lionnet *et al.*, 2007; Valentine *et al.*, 2001), whereas the RecQ is a monomeric helicase from *E. coli* playing a central role in DNA repair (Bagchi *et al.*, 2018; Umezu *et al.*, 1990). For studying DNA rewinding helicases, we combined MT and OT experiments, as done in previous works (Manosas *et al.*, 2013), to explore different force regimes (MT from 5 to 15 pN and OT above 15 pN). In these experiments, we first pulled the extremities of the tethered DNA molecule to partially unzip the hairpin ($\sim 15$ pN). The enzyme's rewinding (or annealing) activity was detected as a decrease in the DNA extension, both in MT and OT assays (Figure 1d and Supplementary Figures S3 and S4). Here, we studied RecG, which is a monomeric helicase from *E. coli*

that promotes the rewinding of DNA strands into duplex DNA (Manosas *et al.*, 2013; McGlynn and Lloyd, 2001). When the rewinding activity is coupled to DNA unwinding, it leads to the formation of a Holliday junction, a four-way DNA structure that is the central intermediate in different DNA repair and recombination pathways (Singleton *et al.*, 2001).

The measured changes in DNA extension can be converted into a number of unwound/rewound bps from the elastic response of the ssDNA (Supplementary Figure S5). This allows us to infer the position of the helicase along the DNA (in bp units) as a function of time, as shown in Figure 3. From these traces, we measured the mean and variance of the helicase displacement and extracted the helicase velocity and diffusivity (insets in Figure 3(a,c,e) and Methods Section *Data analysis*).

### Measuring helicase ssDNA translocation activity

Interestingly, in some cases, we can also monitor the motion of the helicase while it translocates along one strand of ssDNA. In gp41 assays, the experimental traces showed a triangular shape (Figure 1b, upper panel and Supplementary Figure S2), where the rising edge corresponds to the helicase unwinding the hairpin, as previously discussed. After the enzyme reaches the loop and the hairpin has been fully unzipped, the helicase can continue translocating on ssDNA while the hairpin re-anneals in its wake. In this latter process, the DNA hairpin rewinding reaction, observed as a decrease in the measured DNA extension, is limited by the enzyme translocation. We can then infer the position of the helicase as a function of time in the ssDNA translocation process and estimate the helicase velocity and diffusivity, as done with the unwinding traces. For RecQ, the falling edge was not observed (Figure 1c upper panel) because RecQ displays strand switching and repeated unwinding when reaching the loop (Bagchi *et al.*, 2018; Harami *et al.*, 2017). This directed motion towards unwinding precluded the detection of the RecQ ssDNA translocation motion. For RecG, we measured its ssDNA translocation activity by performing experiments at low forces using short oligonucleotides to transiently block the DNA fork (Supplementary Figure S3a and (Manosas *et al.*, 2013)). After the bound oligonucleotide was displaced, the hairpin's rewinding proceeds at a constant velocity, as given by the translocation motion of the helicase. These low-force traces allowed us to measure the RecG velocity and diffusivity on ssDNA.

### Model for helicase movement

The motion of helicase along DNA, driven by the nucleotide hydrolysis reaction, can be described as a random walk on a one-dimensional chain. Translocation is governed by a set of kinetic reactions that connect different helicase-nucleotide states (or conformations) along the DNA chain. The simplest scenario is given by a Poisson model, in which the helicase moves along the DNA in discrete steps of size $d_+$ with exponentially distributed waiting times. The average velocity of the enzyme is given by $v = d_+ / \tau = d_+ k_+$, where $\tau$ and its inverse $k_+$ are the characteristic waiting time and the forward kinetic rate, respectively. The Poisson description is, in general, too simple to capture the dynamics observed in helicases. This is because most helicases exhibit complex mechano-chemical cycles with different rate-limiting steps and multiple pathways. In particular, studies with different helicases have shown the presence of pauses along the helicase trajectories generated by off-pathway states (Dumont *et al.*, 2006;

Ribeck *et al.*, 2010; Seol *et al.*, 2019; Craig *et al.*, 2022). Besides pauses, there are backward steps. In some cases, these backward steps represent intermediate transitions within forward steps (Spies, 2014; Laszlo *et al.*, 2022); in other cases, backward steps reflect slippage events, where the enzyme loses contact with the DNA strand and moves back several bases (Manosas *et al.*, 2010; Seol *et al.*, 2019; Manosas *et al.*, 2012; Schlierf *et al.*, 2019; Sun *et al.*, 2011).

Here we propose a minimal CTRW model that incorporates the key features of helicase movement, including forward and backward steps and pauses. In the CTRW model (Figure 2a), transitions are chosen with a probability $P_+$ to move right, $P_-$ to move left, and $P_0$ to enter the pause state with exponentially distributed times of average $\tau_+$, $\tau_-$ and $\tau_0$, respectively. For simplicity we assume that forward and backward transitions are characterized by constant steps, $d_+$ and $d_-$, respectively. Using the CTRW framework, we can compute the average velocity and diffusivity as a function of the probabilities $P_{+,-,0}$, the transition times $\tau_{+,-,0}$ and the step-sizes $d_{+,-}$, (Eqs. 3 and 4) (Supplementary Section V and Methods Section *CTRW model*). In the context of chemical reactions, kinetic rates $k_i$ are used instead of probabilities $P_i$ and intrinsic transition times $\tau_i$. To express the model in terms of kinetic rates, we consider the following assumptions (Figure 2a): (i) The on-pathway forward reaction and the off-pathway backward (slippage) reaction are irreversible, with rates given by $k_+ = P_+ / \tau_+$ and $k_- = P_- / \tau_-$, respectively; (ii) The off-pathway pause transition is characterized by pause entry and exit rates, given by $k_p = P_0 / \tau_p$ and $k_{-p} = 1 / \tau_{-p}$, with $\tau_0 = \tau_p + \tau_{-p}$; (iii) The intrinsic transition time from the initial state to any other state is assumed to be the same for all transitions $\tau_+ = \tau_- = \tau_p = \tau \ll \tau_{-p} \rightarrow \tau_0 \sim \tau_{-p}$ (Supplementary Section V for details).

The kinetic scheme is shown in Figure 2b and includes three transitions: the ATP-driven forward transition (purple arrow), the slippage backward transition (yellow arrow), and the pausing transition (red arrows). The first two transitions are irreversible and connect different positions of the active helicase state (E) along the DNA track. The third one connects the active helicase state (E) with the pause-inactive one (E*). The ATP-hydrolysis forward transition includes several intermediates as depicted in Figure 2b, but it is described with a single rate, $k_+$, that includes ATP binding-unbinding and ATP hydrolysis. Using the CTRW formalism, we can write the average velocity and diffusivity as a function of the kinetic rates as (Section *CTRW model* in Methods):

$$v = \frac{a_1}{E} \tag{5}$$

$$D = \frac{1}{2E}\left[c_1 + 2\left(\frac{a_1}{E}\right)^2 \frac{k_p}{k_{-p}}\left(\frac{1}{k_p} - \frac{1}{k_t}\right)\right] \tag{6}$$

$$k_t = k_+ + k_- + k_p$$
$$E = \frac{k_+}{k_t} + \frac{k_-}{k_t} + \frac{k_p}{k_{-p}}$$
$$a_1 = k_+ d_+ - k_- d_-$$
$$c_1 = k_+ d_+^2 + k_- d_-^2$$

In general, the different kinetic rates can depend on the concentration of the various reactants (enzyme (E), ATP, ADP, inorganic phosphate $P_i$) and the applied force on the experiment. As discussed in Supplementary Section VIII, a general expression for the rates $k_i$ is given by:

$$k_i = \frac{k_{cat}^i [ATP]}{K_M^i + [ATP]} e^{\frac{x_i^\dagger (F - F_c)}{k_B T}}, \quad \text{un(re)winding}$$

$$k_i = \frac{k_{cat}^i [ATP]}{K_M^i + [ATP]}, \quad \text{translocation} \tag{7}$$

with $i = +, -, p, -p$. Based on a Bell-like model description (Bell, 1978), the rates are exponential with the force $F$ times a transition state distance $x_i^\dagger$, which is related to the change in DNA extension along the kinetic step $i$. The expression (7) assumes that the force only affects the helicase unwinding/rewinding activity, but not the helicase translocation along ssDNA, where we take $x_i^\dagger = 0$. At $F_c \sim 15\text{pN}$, where the hairpin mechanically unzips, the helicase velocity reduces to the translocation velocity that only depends on [ATP]. The ATP dependence is based on a Michaelis–Menten expression (Michaelis *et al.*, 1913) where $k_{cat}^i$ is the rate at ATP saturating conditions and $K_M^i$ is the Michaelis–Menten constant defined as the ATP concentration where the reaction velocity is $k_{cat}/2$. Note that depending on the values of $K_M^i$ and $k_{cat}^i$, the transitions associated with a specific rate $k_i$ would (i) involve ATP hydrolysis (finite $K_M^i$ and $k_{cat}^i$), (ii) involve ATP binding but not hydrolysis ($K_M^i \gg [ATP]$), or (iii) not depend on the ATP ($K_M^i \ll [ATP]$). These different scenarios are explored during the model fitting process (see next section).

### Best-fitting model

The general model proposed (Figure 2b) considers forward and backward steps of size $d_+$ and $d_-$ and four different kinetic rates, $k_+$, $k_-$, $k_p$, and $k_{-p}$. The rates have their [ATP] and force dependence, as described by Eq. (7), through three independent parameters: $K_M$, $k_{cat}$, and $x^\dagger$ summing up to 14 different free parameters. The model includes several simplified cases: the Unidirectional model (Uni-model) without backtracking ($k_- = 0$), the Random walk model (RW-model) without pausing ($k_p = 0$), and the Poisson model described above ($k_- = k_p = 0$), involving 10, 8, and 4 free parameters, respectively. They are schematically shown in Figure 2d.

To reduce the number of free parameters, we analysed the helicase pauses separately. Using a pause detection algorithm (Truong *et al.*, 2020) (Supplementary Section VI and Figure S10), we measure the waiting times to enter and exit pauses, $t_p$ and $t_{-p}$. Both times follow an exponential distribution (insets in Figure 3b,d, f), from which we derive the average time to enter and to exit the pause, $\langle t_p \rangle$ and $\langle t_{-p} \rangle$, and the corresponding rates $k_p = 1/\langle t_p \rangle$ and $k_{-p} = 1/\langle t_{-p} \rangle$ (Section *Data analysis* in Methods). The exponential behaviour agrees with the single-rate limiting step assumption for a single pause state. Lower panels of Figure 4(a,c,e) show $k_p$ and $k_{-p}$ as a function of the applied force for the three enzymes. Rates are exponentially dependent on force as predicted by Eq. (7). The ATP dependence is also well described using the same equation. The overall fit of the force and ATP-dependent rates $k_p$ and $k_{-p}$ to Eq. (7) allows determining the values of $k_{cat}^p$, $K_M^p$, $x_p^\dagger$, $k_{cat}^{-p}$, $K_M^{-p}$ and $x_{-p}^\dagger$, reducing the number of model parameters from 14 to 8 for the general model and from 10 to 4 for the Uni-model. The number of fitting parameters for the RW-model and the Poisson model remains unchanged as they do not consider pauses.

To select the best model that fits the experimental velocity and diffusivity data with the least number of parameters, we performed a least-squares minimization of the reduced chi-square ($\chi_v^2$) value

minus 1, where $v$ stands for the number of degrees of freedom of the fit, equal to the number of data points minus the number of fitting parameters. The goal is to obtain a value of $\chi_v^2$ as close as possible to 1. Values of $\chi_v^2 \gg 1$ indicate a poor model fit, and $\chi_v^2 \ll 1$ indicate over-fitting. We also used the Akaike Information Criterion (AIC) and the Bayesian Information Criterion (BIC) (Akaike, 1974; Schwarz, 1978), that are statistical tools for model selection; they give a numerical value that balances the goodness of the fit with the number of parameters (Supplementary Section VII).

We fitted the general model and the three sub-models (Uni, RW, and Poisson) to the experimentally measured velocity and diffusivity of the three helicases studied: gp41, RecQ, and RecG (Figure 4a,c,e, upper panels). For each helicase and model, we obtained the AIC and BIC values (Table 2), selecting the best-fitting model as the model with the lowest AIC/BIC values ensuring $\chi_v^2 \approx 1$. For the three helicases, the fits to the RW and Poisson models lead to $\chi_v^2 \gg 1$, showing that these models fail to reproduce the experimental data (Table 2 and Supplementary Figure S15). In other words, in the absence of an off-pathway pause state, we cannot reproduce simultaneously the measured helicase velocity and diffusivity.

The best-fitting model for RecQ is the general model. For gp41 and RecG both the Uni-model and the general model fit the data with similar values of $\chi_v^2$, AIC and BIC (less than 10% differences in their values, Table 2). Note that the difference between the two models is that the general model includes a backward slippage pathway, whereas the Uni-model does not. Helicase slippage has been previously observed for gp41 (Manosas *et al.*, 2010; Manosas *et al.*, 2012), RecQ (Seol *et al.*, 2019), and other helicases (Schlierf *et al.*, 2019; Sun *et al.*, 2011). Moreover, large slippage events (> than 10 bp) are observed in our experimental traces for the three helicases (Supplementary Figure S18), but they are not included in the velocity and diffusivity analysis. However, smaller slippage events might be masked in the experimental signal. As a consequence, we choose the general model for the three helicases.

Once the model had been selected, we performed a second optimization step for each helicase type by identifying weak dependencies on force and [ATP] of the rates $k_+$, $k_-$, $k_p$, $k_{-p}$ in Eq. (7) Table 1. We have tested $x^\dagger = 0$ for the rates that weakly depend on force and $K_M = 0$ for the rates that weakly depend on concentration, further reducing the number of parameters. In Table 2 and Figure 4, we show, for the three helicases, the results from the overall optimization process that leads to the best model with the minimum number of parameters. For RecQ we only used unwinding data, whereas for gp41 and RecG we simultaneously fitted the average velocity and diffusivity using un(re)winding and ssDNA translocation data. Interestingly, for the ssDNA translocation activity, the measured velocity and diffusivity are independent of force for both helicases (Figure 4, grey shaded area Table 1). This finding supports the view that the main role of the mechanical force is altering the duplex stability, therefore affecting the DNA un(re) winding reaction but not the enzyme translocation along ssDNA.

For all helicases, the forward rate $k_+$ depends on [ATP], with its Michaelis–Menten constant, $K_M^+$, very close to the $K_M$ obtained by fitting the average un(re)winding velocity as a function of [ATP] using the Michaelis–Menten expression: $v = \frac{k_{cat}[ATP]}{K_M + [ATP]}$ (Supplementary Figure S16 and Table 3). This shows that the main ATP dependence comes from the on-pathway ATP hydrolysis coupled with the translocating forward step. In contrast, the force dependence is different for each helicase. For RecQ and RecG, $k_+$ is weakly dependent on force, whereas for the gp41,

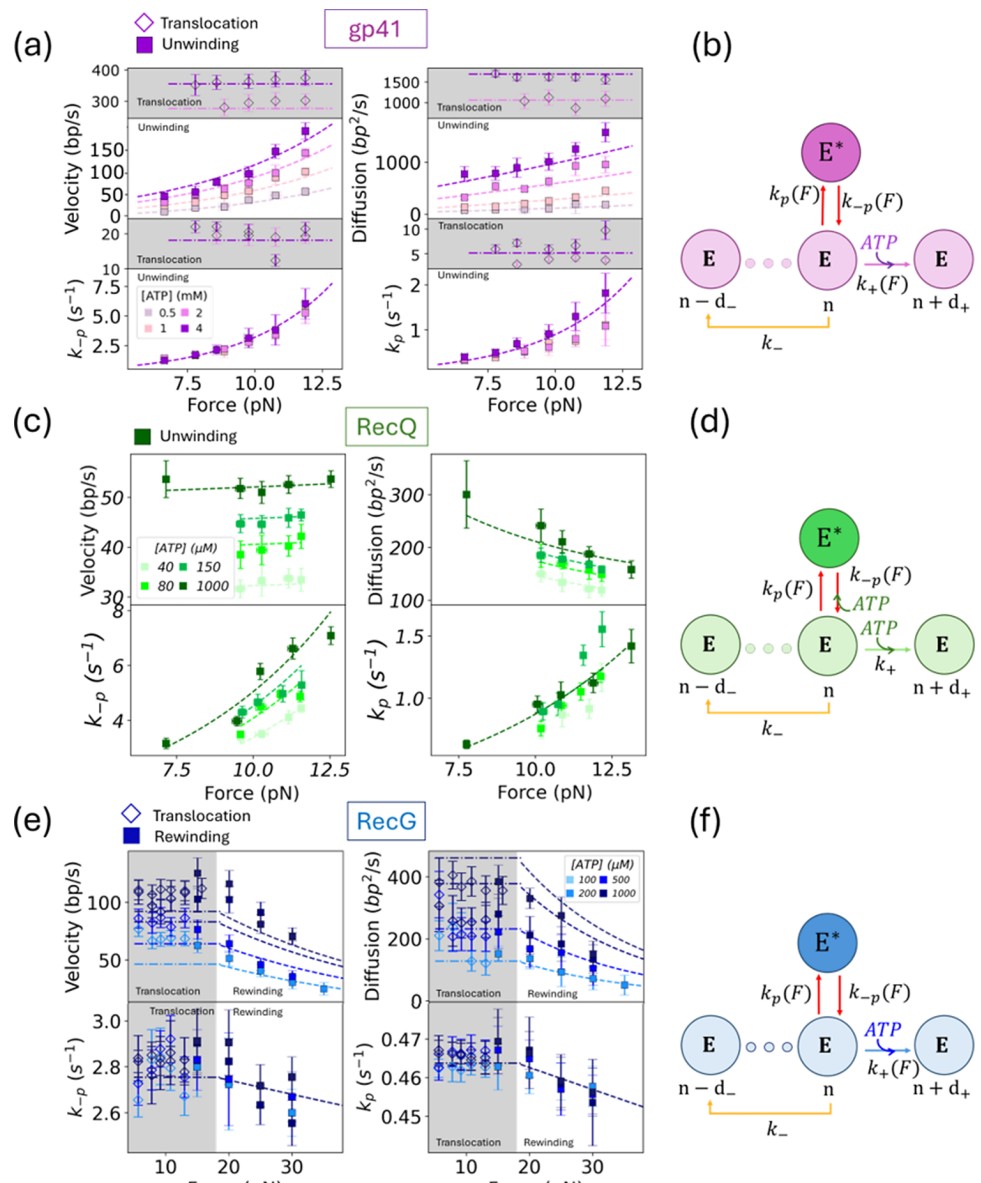

**Figure 4.** Best-fitting models (a,c,e). The measured velocity (top left), diffusivity (top right), exit pause rate (bottom left), and entry pause rate (bottom right) as a function of force at different ATP conditions for gp41 (purple), RecQ (green), and RecG (blue). The filled squares and empty diamonds are computed from the un(re)winding and translocation data, respectively. Values shown are the mean between different molecules and the error bars represent the standard error of the mean. For gp41, we average ∼20 beads with ∼20 traces for each bead, for RecQ ∼10 beads with ∼10 traces each, and for RecG ∼10 beads with ∼10 traces each. The dashed lines are the best fit to the model, as given by Eqs. (5, 6). (b,d,f) Schematic representation of the best-fitting model, showing the ATP and force dependence of the different kinetic rates.

**Table 1.** Force and ATP dependencies of the rates involved in equation S.17 for the three studied helicases

|  | $k_+$ | $k_-$ | $k_p$ | $k_{-p}$ |
|---|---|---|---|---|
| gp41 | *F*: **Yes** | *F*: **No** | *F*: **Yes** | *F*: **Yes** |
|  | [ATP]: **Yes** | [ATP]: **No** | [ATP]: **Yes** | [ATP]: **No** |
| RecQ | *F*: **No** | *F*: **No** | *F*: **Yes** | *F*: **Yes** |
|  | [ATP]: **Yes** | [ATP]: **No** | [ATP]: **No** | [ATP]: **Yes** |
| RecG | *F*: **Yes** | *F*: **No** | *F*: **Yes** | *F*: **Yes** |
|  | [ATP]: **Yes** | [ATP]: **No** | [ATP]: **No** | [ATP]: **No** |

*Note:* A schematic diagram is shown in Figure 4(b,d,f) with the corresponding dependencies.

**Table 2.** Comparison of models using $\chi_v^2$, AIC, and BIC

| Enzyme |  | gp41 |  |  | RecQ |  |  | RecG |  |
|---|---|---|---|---|---|---|---|---|---|
| Estimator | $\chi_v^2$ | AIC | BIC | $\chi_v^2$ | AIC | BIC | $\chi_v^2$ | AIC | BIC |
| General | **1.77** | **999** | **1013** | **1.32** | **24** | **35** | **2.12** | **1137** | **1158** |
| Uni | 2.01 | 1053 | 1062 | 10 | 158 | 165 | 2.62 | 1140 | 1162 |
| RW | 8 | 1309 | 1331 | 9 | 82 | 94 | 5 | 1298 | 1210 |
| Poisson | 16 | 1493 | 1502 | 11 | 165 | 173 | 27 | 1526 | 1435 |

*Note:* In bold, we marked the best values of the estimators.

**Table 3.** Parameters obtained from fitting un(re)wnding and translocation data shown in Figure 4 using the best-fitting model protocol described in Section *Best-fitting model*

| Parameter | gp41 ($d_+ = 1$ bp) | Error | RecQ ($d_+ = 1$ bp) | Error | RecG ($d_+ = 3$ bp) | Error |
|---|---|---|---|---|---|---|
| $k_{cat}^+$ (s$^{-1}$) | 644 | 5 | 101 | 3 | 33 | 1 |
| $K_M^+$ ($\mu$M) | 1700 | 50 | 20 | 3 | 110 | 21 |
| $x_+$ (pm) | 190 | 10 | - | - | $-30$ | 1 |
| $k_{cat}^-$ (s$^{-1}$) | 4.1 | 0.9 | 4.9 | 0.3 | 1.4 | 0.1 |
| $K_M^-$ (mM) | - | - | - | - | - | - |
| $x_-$ (pm) | - | - | - | - | - | - |
| $k_{cat}^p$ (s$^{-1}$) | 5.08 | 0.01 | 1.71 | 0.02 | 0.46 | 0.01 |
| $K_M^p$ (mM) | - | - | - | - | - | - |
| $x_p$ (pm) | 305 | 11 | 163 | 5 | $-2.0$ | 0.2 |
| $k_{cat}^{-p}$ (s$^{-1}$) | 18.13 | 0.04 | 9.15 | 0.07 | 2.75 | 0.08 |
| $K_M^{-p}$ (mM) | - | - | 0.02 | 0.08 | - | - |
| $x_{-p}$ (pm) | 312 | 3 | 165 | 12 | $-1.9$ | 0.1 |
| $d_-$ (bp) | 4 | 1 | 4 | 1 | 2 | 1 |

*Note:* Errors extracted from the standard deviation of the parameters obtained performing the fitting several times.

$k_+$ markedly increases with force (Table 3), in agreement with the reported active and passive character of these helicases, respectively (Manosas *et al.*, 2010; Manosas *et al.*, 2013). On the other hand, the pause kinetics have specific ATP and force dependencies for each helicase. In particular, the pause kinetics only depend on the ATP concentration for the RecQ case, in agreement with previous measurements (Seol *et al.*, 2019). An ATP-dependent pause kinetics has been observed in other enzymes (Dumont *et al.*, 2006; Burnham *et al.*, 2019). Mechanistically, ATP-dependent transitions into or out of paused states may reflect conformational rearrangements of the helicase that require nucleotide binding or hydrolysis (Ali and Lohman, 1997; Rudolph and Klostermeier, 2015; Theissen *et al.*, 2008).

The model can be fitted to the experimental data (velocity and diffusivity) with similar values of $\chi_v^2 \approx 1$ using a large range of $k_+$ and $d_+$ values, which are almost inversely correlated. To limit the range of step-sizes, we explored different values around each AIC/BIC minimum and identified a spectrum of values compatible with a relative difference of less than 5% in both AIC and BIC. This analysis led to step-sizes of $d_+ = 1 - 3$ bp for gp41, $d_+ = 0.5 - 1$ bp for RecQ, and $d_+ = 3 - 4$ bp for RecG. Interestingly, these values are in agreement with previously estimated step-sizes for these helicases: $d_+ = 1$ bp for gp41 or other hexameric helicases (Lionnet *et al.*, 2007; Schlierf *et al.*, 2019; Pandey and Patel, 2014); $d_+ = 1$ bp for RecQ (Craig *et al.*, 2022; Sarlós *et al.*, 2012) and bps $d_+ = 2 - 4$ for RecG (Manosas *et al.*, 2013; Martinez-Senac and Webb, 2005; Toseland *et al.*, 2012). Accordingly, we have chosen $d_+ = 1$ bp for gp41 and RecQ, and $d_+ = 3$ bp for RecG. Finally, the backward slippage transition is described with a force and ATP-independent rate $k_-$ and a backward step $d_- \sim 2 - 4$ bps for the three helicases. Recent studies suggest that some helicases might display a variable step-size (Ma *et al.*, 2020). On the other hand, helicase slippage occurs along a random number of nt in ATP and force-dependent manner ((Manosas *et al.*, 2012) and Supplementary Figure S18). Therefore, the model could be refined by considering variable forward and step-sizes $d_+$ and $d_-$, with force and ATP dependencies.

## On the efficiency of helicases

The trade-off between the energy cost and the efficiency of helicases can be investigated through the thermodynamic uncertainty relation (TUR) (Song and Hyeon, 2021). The TUR is an inequality relating the uncertainty (or precision) in the motor activity and the energy from ATP hydrolysis that is irreversibly lost to the environment as heat $q$, known as the entropy production rate $\sigma = \dot{q}/T$. For an arbitrary current $\dot{x}$ in a nonequilibrium steady state, the time-integrated current $X(t) = \int_0^t \dot{x}(s)\,ds = x(t) - x(0)$ satisfies the TUR inequality (Pietzonka *et al.*, 2017):

$$\sigma \geq \frac{2\langle X(t)\rangle^2}{V_{X(t)}t}k_B, \tag{8}$$

where $\langle X(t)\rangle$ and $V_{X(t)} = \langle X(t)^2\rangle - \langle X(t)\rangle^2$ are the mean and variance of the integrated current measured during a time interval $t$. From Eq. (8) one can define the dimensionless $\mathcal{Q}$ factor that quantifies the tightness of the TUR inequality (Song and Hyeon, 2021),

$$\mathcal{Q} = \frac{\sigma V_{X(t)}}{k_B\langle X(t)\rangle^2}t \geq 2. \tag{9}$$

For helicases, $X$ is the motor displacement measured by the bead's position. From the velocity $v = \frac{\langle X(t)\rangle}{t}$ and diffusivity $D = \frac{V_{X(t)}}{2t}$ we get,

$$\mathcal{Q} = \sigma\frac{2D}{k_B v^2} \geq 2. \tag{10}$$

Previous studies have shown that, for translocating motors, the TUR bound is loose with $\mathcal{Q}$ values ranging from 5 to 20 for kinesin, from 5 to 13 for myosin, or from 50 to 100 for T7 DNA polymerase and the ribosome (Song and Hyeon, 2021; Hwang and Hyeon, 2018; Song and Hyeon, 2020; Piñeros and Tlusty, 2020). However, to our knowledge, $\mathcal{Q}$ has never been investigated for helicases.

To determine $\mathcal{Q}$, we measure $v$ and $D$ from the time traces of the motor. In addition, $\sigma$ can be estimated assuming a tight mechano-chemical coupling between the unwinding-rewinding of $d_+$ bps and the hydrolysis of one ATP (Sun *et al.*, 2011; Xie, 2020). Moreover, we assume that ATP is not consumed in the backward and pausing steps. If $q$ is the heat per step irreversibly lost to the environment, $\sigma$ can be written as: $\sigma = \frac{q}{T}\frac{v}{d_+}$ with $d_+ = 1$bp for gp41 and RecQ and $d_+ = 3$bp for RecG. Besides the heat $q$, the energy balance contains the chemical ($\Delta\mu$) and mechanical contributions ($W$), $q = \Delta\mu - W$, where $W = d_+\left(\Delta G_{bp} + W_F\right)$ is the reversible mechanical work needed to unzip or rezip $d_+$ bps at force $F$ (Supplementary Figure S17 and Section IX). $\Delta G_{bp}$ is the hybridization or melting free energy per bp, $\Delta G_{bp} \approx 2k_BT$ (Huguet *et al.*, 2010), and $W_F$ is the stretching contribution at force $F$, which can be estimated using elastic models for the ssDNA polymer (Supplementary Section IX). Finally, the energy released from ATP hydrolysis $\Delta\mu \approx 12 - 20k_BT$, depending on the ATP concentration.

The $\mathcal{Q}$ factor is related to the thermodynamic efficiency. The second law implies $\sigma \geq 0$, and therefore $q \geq 0$ or $W \leq \Delta\mu$. We define the motor efficiency $\eta$ as the ratio between the amount of mechanical work per step $W$ and the available chemical energy from ATP hydrolysis $\Delta\mu$, $\eta = \frac{W}{\Delta\mu}$. Using the energy balance, $q = \Delta\mu - W$ we can write the efficiency as,

$$\eta = \frac{W}{\Delta\mu} = \frac{1}{1 + q/W} = \frac{1}{1 + \frac{\sigma T d_+}{vW}}, \quad (11)$$

For the case of a molecular motor un(re)winding DNA, $\mathcal{Q}$ is proportional to $\sigma$ (Eq. 10) and $\eta$ can be written as (Song and Hyeon, 2021; Pietzonka *et al.*, 2016):

$$\eta = \frac{1}{1 + \mathcal{Q}\frac{vd_+}{2D}\frac{k_BT}{W}}. \quad (12)$$

To calculate $\mathcal{Q}$ and $\eta$ we use Eqs. (10 and 11) using the measured values of $v, D$ and the estimated $\sigma$. In Figure 5, we show $\mathcal{Q}$ and $\eta$ for the three studied helicases in a log–log scale, as a function of the ATP concentration at different forces. RecQ and gp41 present large $\mathcal{Q}$ values $100 - 200$ and $200 - 500$, respectively and low efficiencies at zero force around 0.1, which further decrease with force. In contrast, RecG has lower $\mathcal{Q}$ values $\sim 10 - 20$, which decrease upon increasing the force above $F_c$. Interestingly, lower $\mathcal{Q}$ values correlate with higher $\eta$, with RecG reaching an efficiency close to 1 at the stalling force of $\sim 35 - 40$pN (Manosas *et al.*, 2013). In the inset of Figure 5b, we plot $\eta$ versus $\mathcal{Q}$ in a linear-log scale. While gp41 and RecQ fall at the bottom right inefficiency corner of high $\mathcal{Q}$ -low $\eta$ values, RecG follows a trend with $\eta$ increasing upon decreasing $\mathcal{Q}$, approaching its maximum $\eta = 1$ if $\mathcal{Q} \rightarrow 2$. A two-parameter fit to the function $\eta = 1 + a\log(\mathcal{Q}/2) + b\log^2(\mathcal{Q}/2)$ gives $a = 0.12, b = -0.17$ (inset, continuous black line). The significance of this fit lies in the logarithmic dependence of $\eta$ with $\mathcal{Q}$, underlying a fundamental looseness of the TUR regarding the thermodynamic efficiency of molecular machines. Comparing the unwinding helicases, RecQ, and gp41, we observe that $\mathcal{Q}$ depends on the passive and active nature of the enzymes with larger $\mathcal{Q}$ values for passive helicases, yet $\eta$ remains qualitatively similar. This is due to the fact that the helicase mechanism (passive versus active) affects the values of $D$ and $v$ (Supplementary Figure S7a), and consequently the value of the $\mathcal{Q}$ factor. However, the balance between the energy available

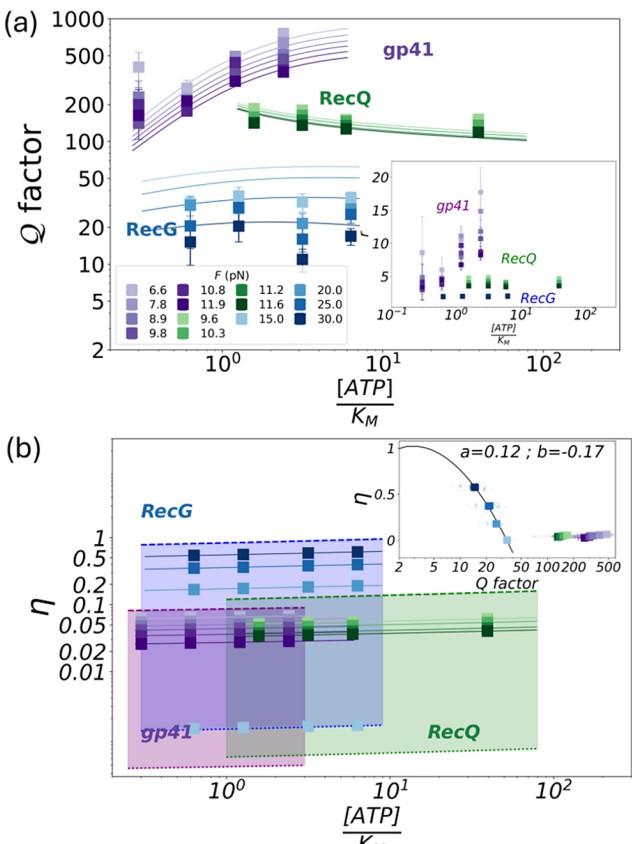

**Figure 5.** (a) Estimated $\mathcal{Q}$ factor as a function of ATP concentration for different forces in a log–log scale for the three studied helicases, gp41 in purple, RecQ in green, and RecG in blue. The ATP concentration is divided by the Michaelis–Menten constant of each helicase. Inset shows the randomness parameter as a function of the ATP, the effect of pauses (b) Efficiency $\eta$ as a function of the ATP concentration for different forces in a log–log scale. The inset shows $\eta$ as a function of $\mathcal{Q}$ for the three helicases on a linear-log scale. The continuous line shows the fit to the RecG data of the function $\eta = 1 + a\log(\mathcal{Q}/2) + b\log^2(\mathcal{Q}/2)$.

from ATP hydrolysis and the work, which governs $\eta$, does not explicitly depend on $D$ and $v$: $\eta = \frac{W}{\Delta\mu} = \frac{d_+\left(\Delta G_{bp} + W_F\right)}{\Delta\mu}$. Indeed, at a given force $F$ and ATP concentration, $\eta$ is mainly governed by the helicase step-size, with larger step-sizes leading to higher $\eta$ values.

From the velocity and diffusivity measurements we can also estimate the random parameter (Svoboda *et al.*, 1994), which, in simple cases, is related to the number of rate-limiting steps in an enzymatic cycle. It is defined as $r = \frac{2D}{dv} = \frac{\mathcal{Q}k_BT}{q}$. The results for $r$ are shown in the inset of Figure 5a for the different enzymes and different experimental conditions. It is interesting to note that, in general, $r$ is larger than one, which is associated with increased motor diffusivity. Both DNA sequence heterogeneity and the presence of pauses can lead to large $D$ values (Shaevitz *et al.*, 2005). In order to investigate how the interplay between DNA sequence and pauses affects the helicase diffusivity, we have performed simulations of the CTRW model including the DNA sequence of the hairpin substrate. The results show that, for the helicases studied, the effect of pauses largely predominates over the DNA sequence effects. As shown in the Supplementary Section IV, the diffusivity $D$ increases strongly in the presence of pauses, whereas the DNA sequence does not significantly affect its value (Supplementary Figure S7b).

## Discussion and conclusions

DNA replication and repair are fundamental processes of life by which genetic information is preserved and transferred to the next generation. These processes require the action of helicases that use ATP hydrolysis to move on DNA, inducing the unwinding and rewinding of the double helix (Lohman and Bjornson, 1996; Tuteja and Tuteja, 2004; Delagoutte and Von Hippel, 2003; Pyle, 2008). In this work, we use magnetic and optical traps to monitor the motion of different DNA helicases (gp41, RecQ, and RecG), while they move along DNA under different forces and ATP concentrations (Figure 1).

Helicase dynamics can be characterized by measuring the velocity when the enzyme catalyses the unwinding or rewinding reaction and the velocity when it translocates along a single DNA strand. As shown in previous works, analysing how these velocities vary with applied force and ATP concentration provides insight into the helicase mechano-chemical cycle (Manosas *et al.*, 2010; Ribeck *et al.*, 2010; Spies, 2014; Seol *et al.*, 2019; Laszlo *et al.*, 2022; Manosas *et al.*, 2013; Lionnet *et al.*, 2007). Here, we extend this analysis by measuring the helicase diffusivity, which allows us to better characterize the helicase dynamics (Figure 3). We use a Continuous Time Random Walk (CTRW) model, depicted in Figure 2, to describe the helicase dynamics that include forward and backward steps and pauses. Analytical expressions for the helicase velocity and diffusivity can be derived and used to fit the data, allowing us to infer the minimal number of kinetic states and transitions necessary to capture the observed dynamics for each helicase (Figure 4). As shown in a recent work (Burnham *et al.*, 2019), an analysis of the first-passage-time distribution can also be used to extract forward and backward rates and pause kinetics of motors. However, the study of motor diffusivity presented here also allows us to explore fundamental thermodynamic constraints using the $Q \geq 2$ factor of the Thermodynamic Uncertainty Relation (TUR), which is related to the helicase efficiency.

Importantly, incorporating an off-pathway pause state into the CTRW model is essential to reproduce the experimental data for all three helicases studied. Models without pauses, such as the Poisson or Random Walk model, do not fit the data, raising the question of the biological role of pauses. Helicases work in coordination with other enzymes to perform their biological functions. The RecQ and RecG helicases work with single-stranded binding proteins and other accessory proteins in different DNA repair pathways. Gp41 operates as part of a large complex containing two polymerases and other proteins, known as the replisome and is responsible for replicating the genomic DNA in T4 bacteriophage. Previously, we have shown that when the gp41 helicase works together with the polymerase, the velocity of the helicase advance increases without pausing (Manosas *et al.*, 2012). Therefore, pausing might be the strategy to control helicase activity. Without the accessory proteins needed to develop a specific biological function, such as replication and repair, pauses stall the helicase activity.

A general feature observed for all helicases is that force affects unwinding and rewinding activity but not the translocation activity along one strand of DNA (Figure 4a,e, white versus gray background). The main effect of force is to destabilize the DNA duplex in a helicase-dependent manner. For gp41, the velocity and diffusivity are very sensitive to the value of the applied force, whereas for RecQ and RecG, they are not. Indeed, when the force changes by 5 pN, the velocity and diffusivity change by a factor of 10 for gp41, whereas they remain almost constant for RecQ and RecG

(Figure 4a,c,e). This force sensitivity is related to their active and passive character, as discussed elsewhere (Manosas *et al.*, 2010).

Assuming a tight mechano-chemical coupling and using the diffusivity measurements, we estimate the thermodynamic uncertainty factor $Q$ and the efficiency $\eta$. RecG presents the smallest $Q$ factor and largest efficiency as compared to gp41 and RecQ (Figure 5). In particular, at forces close to the stalling force $\sim 40$ pN, RecG reaches $\eta \sim 1$ by operating close to the thermodynamic optimization limit, $Q = 2$. This large efficiency correlates with its large step size of 3 bp. In contrast, gp41 and RecQ, which unwind only one bp per ATP hydrolysed, present much lower $\eta$, below 0.15. Remarkably, only RecG during rewinding shows efficiencies approaching 1, whereas unwinding activity for gp41 and RecQ helicases is thermodynamically inefficient, with most of the energy from ATP hydrolysis released as heat. This fact indicates that the efficiency of molecular machines is largest whenever they operate uphill in the sense that the energy cost of the task $W$ is comparable to the chemical energy from ATP hydrolysis, $\Delta\mu$. A similar phenomenon occurs for F1 Fo ATPase (Yasuda *et al.*, 1998), which is almost 100% efficient when transporting protons against the electrochemical potential gradient to synthesize ATP. In contrast, in the presence of thermogenic proteins, ATP synthase decouples from the proton gradient, and the proton flow is employed to produce heat.

The CTRW framework proposed in this work allows for characterization of helicase dynamics through velocity and diffusivity measurements, which can be directly obtained from single-molecule assays. The model proposed can be adapted to describe different types of molecular motors that move along DNA or through other templates, such as polymerases or kinesins, and it can be easily extended to motors that have variable step-sizes or multiple pause states.

**Open peer review.** To view the open peer review materials for this article, please visit http://doi.org/10.1017/qrd.2025.10011.

**Supplementary material.** The supplementary material for this article can be found at http://doi.org/10.1017/qrd.2025.10011.

**Acknowledgements.** V.R-F., M.M., and F.R. acknowledge support from the Spanish Research Council Grant [PID2022-139913NB-I00]; M.M. acknowledges support from MICIU/AEI/10.13039/501100011033 and NextGenerationEU/ PRTR [CNS2022-135910]. Work in the Bianco laboratory is supported by NIH Grant GM144414 to PRB. F.R. also acknowledges support from the ICREA Academia Prizes 2018 and 2023.

**Competing interests.** The authors declare none.

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
