## [Reviewer Report]

The authors describe a rigorous single-molecule study of three helicase: gp41, RecQ, and RecG, during unwinding or rewinding of DNA hairpins using a combination of low resolution optical tweezers and magnetic tweezers. While other helicases (e.g. XPD, Rep, RecBCD) has been studied using other single molecules methods such as TIRF, smFRET, and high-resolution optical tweezers, is to the reviewer’s knowledge that the study of mechanochemical cycle of gp41, RecQ, and RecG is entirely new. Nonetheless, is to the reviewer’s opinion that the major novelty of the research is the analysis and model design to discriminate among possible mechanisms of action of helicases. Therefore, this type of analysis can be performed to other molecular motors studied with single molecule methods. My specific comments are delineated bellow.

1. Why do the authors used those specific concentrations of helicases? Please, mention if this value is based on the dissociation constant or something else.

2. How do the authors prevent the binding of more than one helicase to their respective hairpin?

3. Any reason why the activity of gp41 has a wide distribution of velocities? Some traces looks like go twice faster than others. The activities of RecQ and RecG are more homogenous. The activities of gp41 also look noisier than the other helicases. Why?

4. In figure 3b, 3d and 3f. It looks like the pause identification in is not optimal in these representative traces. For instance, if the shortest red region is a pause in Figure 3f, there are similar or even larger pausing regions no identified. Aso, if in figure 3b around 7 sec there is a short pause, then the full trace is basically a collection multiple pauses. Maybe the authors can be more rigorous in the definition of a pause. How can this affect the analysis of the data and discrimination of the model of action of helicases? Can the authors play with different thresholds to identify pauses and re-do the analysis to show that the basic model of action of helicases does not change? Finally, what exactly is a pause? why is it important besides being found during activity?

5. To validate their identification of pauses the authors performed simulation and they claim that their algorithm can identify pauses with an accuracy of 5 %. However, it is clear to the reviewer that this is not the case by looking at the representative traces of figure 3. Maybe, one explanation is that the CTRW simulation accounts for the thermal noise but does not consider the instrumental noise such as the one introduced by the bead attached to the micropipette in the low resolution optical tweezers.

6. How do the authors know if some “pauses” are instead events where the helicase gets release, another helicase binds and resumes the activity?

---

## [Reviewer Report]

Franco and co-authors apply a continuous time random walk (CTRW) model formalism to single-molecule helicase DNA unwinding, rewinding, and translocation measurements of three helicases (gp41, RecG, and RecQ). By measuring these rates, off-pathway pause states, and the variance in the rates over time, as a function of ATP concentration and applied force, the authors obtain muti-parameter data sets that are suitable for globally fitting to different implementations of the CTRW model. By optimizing the fitting, the authors obtain detailed kinetic models for the activity of the three helicases. The authors extend the analysis to consider the efficiency of converting the energy of ATP hydrolysis to mechanical work and excess heat though the application of the previously developed thermodynamic uncertainty relation formalism.

The approach the authors have taken to model the helicase kinetics is novel and the characterization of the activity of the three helicases as a function of ATP concentration and force is an valuable contribution to the field. The CTRW modeling approach provides intriguing results and, baring some specific points described in detail below, offers a possibly powerful approach to model helicase activity. Extending the modeling results to address the efficiency and thermodynamic uncertainty relation of the helicases provides novel insights into helicase activity that may have broader physiological or mechanistic ramifications. Overall I favor publication of this extensive and interesting work once the authors have addressed the following important points.

1. Page 2– the definition of the randomness parameter in terms of the diffusivity parameter, D, may be somewhat misleading. Although the units are equivalent to diffusivity the randomness reflects the statistical mechanics of the enzymatic process and the related D parameter reflects these details, rather than diffusion in the well-understood meaning. For example, inherent variability in the substrate has been shown to increase the randomness parameter above 1 ( Statistical Kinetics of Macromolecular Dynamics (2005) Shaevitz et al. Biophysical Journal). On this point, the approaches developed in the Shaevitz work, which are an extension and generalization of the cited work (ref 30), would be highly relevant to the results and approaches developed in this work.

2. Page 2 - A minor technical point, but different purification approaches can result in RecQ maintaining or loosing its HRDC domain that can dramatically influence the activity of the enzyme. It would be worth indicating if the RecQ helicase is WT or HRDC truncated as is frequently the case with some purification schemes.

3. Page 8. It seems that the model assumptions embodied in equation 7 include the possibility that both the pausing entry and exit could depend on the ATP concentration in a Michaelis-Menton manner. This is an unusual assumption that should be explained and rationalized in greater detail. In particular, some physical basis for this assumption should be provided and a discussion of how pause entry and/or exit could depend on the ATP concentration and require ATP catalysis. This would in effect be postulating parallel ATP hydrolysis pathways that would represent a radical new scheme for helicases. Some discussion of the underlying physical interpretations of these assumptions made in the model are required and ideally these should be supported by additional evidence form the existing literature or from additional analysis of the current data.

4. I would suggest adding one additional model in which pausing is dependent on the force but not on ATP – this would provide a test of the proposal that entry into or out of a paused state would depend on ATP. This is a slight modification of the full model that does not include ATP processes associated with pausing.

5. One additional point concerning the CTWR modeling. There is evidence in the literature that some helicases, e.g. the hepatitis C virus NS3 helicase, exhibit a rate limiting step associated with multiple basepair unwinding events, yet they unwind one basepair per ATP hydrolyzed. Could this type of discontinuous behavior be captured by the CTRW approach?

6. Page 10 – analyzing the pause frequency tp makes an assumption that the pausing is occurring with constant kinetics, whereas the kinetics could reflect a probability of pausing per kinetic step, which would be coupled with ATP hydrolysis and therefore the ATP concentration. It is possible that this subtle difference does not impact the analysis but it would be worth discussing this possibility and explaining the choice of defining the pausing entry in time rather than in kinetic cycles or nucleotides unwound or translocated.

7. Page 10 – I understand the rational in measuring as many of the model parameters as possible to increase the accuracy and stability of the model fitting, however, ideally at some step all the model parameters would be simulated and compared with the measured data including pausing kinetics to ensure that the model faithfully captures the measured kinetics. There should be some modeled and analyzed trajectories in the supplemental information closing the cycle by showing that the input parameters are faithfully recovered when simulated with the CTRW model with the appropriate added noise and sampling rate and averaging applied.

8. Page 12 figure 4. The Kp and K-p rates should include units on the graphs. Although the authors mention the relation between the “diffusion” D and the randomness parameter, they do not provide a measure of the randomness. It would be helpful to include the randomness plots in Figure 4 or in a supplemental figure. Can the authors provide a physical explanation for the decrease in the D parameter for RecQ with increasing force despite the fact that the pausing frequency is increasing?

9. Page 13 – Table II. The units on the pausing Kcat values should not be bp/s I think – rather simply 1/s. Can the authors provide a measure of the uncertainty of the parameters? The bac-stepping or back-sliding distances are significant and occur about as frequently as pauses – is there experimental evidence for these back ward steps occurring with the predicted frequency and extent?

10. Page 13. Discussion – the authors are proposing a radical new model for helicase unwinding and translocation – postulating the existence of pause states that are associated with the hydrolysis of ATP – this is a bold and provocative assertion that is not addressed at all in the discussion. These results are somewhat glossed over but they should be discussed and expanded on in the discussion. Are there similar examples from the literature? Is there any supporting evidence for these findings? Can the authors provide physical mechanisms that could give rise to the various ATP and force dependent kinetics of entering into and exiting from pauses?

Minor points

11. The word “lineal” is frequently used instead of “linear”

---

## [Reviewer Report]

The authors have largely addressed my substantial concerns with the initial submission. I appreciate the care and effort the authors put into responding to the critiques of the two reviewers. I support acceptance of the revised manuscript.

there are two small errors that should be corrected prior to publication:

Figure s7: “Note that D/v is lower for passive helicases, indicating a lower Q factor” - the data in the figure suggest the opposite, that active helicases show a lower D/v value. there is a discrepancy somewhere that should be addressed.

supplemental information P17, there is discussion of figure S8 and in particular S8 (c) which does not exist. i am guessing that this is due to adding additional figures, but I could not find the figure corresponding directly to the discussion surrounding Fig S8.